# Bicarbonate-binding catalysis for the enantioselective desymmetrization of keto sulfonium salts

José Alemán [1,2,3] ✉, Jorge Humbrías-Martín[1], Roberto del Río-Rodríguez [1], Fernando Aguilar-Galindo [2,4], Sergio Díaz-Tendero [2,4,5] & Jose A. Fernández-Salas [1,2] ✉

Herein, an enantioselective desymmetrization of cyclic keto sulfonium salts through enantioselective deprotonation/ring opening process by anion-binding catalysis is presented. We report a squaramide/$HCO_3^-$ complex as catalytic active species which is able to stereo-differentiate two enantiomeric protons, triggering the ring opening event taking advantage of the great tendency of sulfonium salts to act as leaving groups. Thus, this desymmetrization methodology give rise to β-methylsulfenylated sulfa-Michael addition type products with excellent yields and very good enantioselectivities. The bifunctional organocatalyst has been demonstrated to be capable of activating simultaneously the base and the keto sulfonium salt by DFT calculations and experimental proofs.

The synthesis of enantioenriched molecules is a central research area in organic synthesis and the development of new methodologies for their preparation still stands as a crucial role for the synthesis of recognizable structures in natural and synthetic biologically active compounds[1–3]. Consequently, a plethora of methodologies and strategies have been developed in the last decades in order to introduce chirality[1,3–5]. In this regard, organocatalysis appeared as an alternative in asymmetric synthesis[6,7]. In this context, nature has served as an inspiration source to chemists considering the advent and development of new methodologies applying novel modes of interaction and activation[8]. Although anion-binding processes are well-known in molecular recognition, this type of activation has only recently been exploited for catalysis[8,9]. Considering the use of an organocatalyst for this purpose, this strategy relies on the ability to recognize/activate negatively charged species with a chiral catalyst bearing a H-bond donor (HBD) core unit, giving rise to the named chiral contact ion pair complex[8,9] through non-covalent interactions. Pioneer studies established that an anionic component of an ion pair may be bonded by a catalyst through hydrogen bond, generating then new reactive species[10,11]. Based on these findings, various research groups have developed different catalytic systems based on this activation mode.

One of the initial approaches, more deeply described in the literature, were focused on mimicking the ability of large complex molecules and enzymes to recognize halide anions via hydrogen bonds[8,9,12,13]. In fact, considering chloride anion recognition, more than 900 X-ray[14] structures of enzyme-chloride interaction systems have been reported in the literature (top-left, Fig. 1)[15]. In this sense, different chiral catalysts with the capability to perform a HBD-halide interaction and promote the asymmetric bond forming event with the corresponding cationic species of the ion pair have been described[12,16]. Molecular recognition of sulfonate anion has served as a source of inspiration, based again in the large number of biological systems that present this kind of interaction (>110 examples)[14] (medium-above, Fig. 1)[17]. Jacobsen´s group has recently reported a squaramide-based bifunctional organocatalyst system that activates the sulfonate anion[18,19]. During the last years, chiral HBD-based catalysts have shown the ability to bind a variety of anions associated with cationic organic intermediates to produce chiral ion pairs susceptible to promote

[1]Departamento de Química Orgánica (módulo 1), Universidad Autónoma de Madrid, Cantoblanco, Madrid, Spain. [2]Institute for Advanced Research in Chemical Sciences (IAdChem), Universidad Autónoma de Madrid, Madrid, Spain. [3]Center for Innovation in Advanced Chemistry (ORFEO-CINQA), Universidad Autónoma de Madrid, Madrid, Spain. [4]Departamento de Química, Universidad Autónoma de Madrid, Cantoblanco, Madrid, Spain. [5]Condensed Matter Physics Center (IFIMAC), Universidad Autónoma de Madrid, Madrid, Spain. ✉e-mail: jose.aleman@uam.es; j.fernandez@uam.es

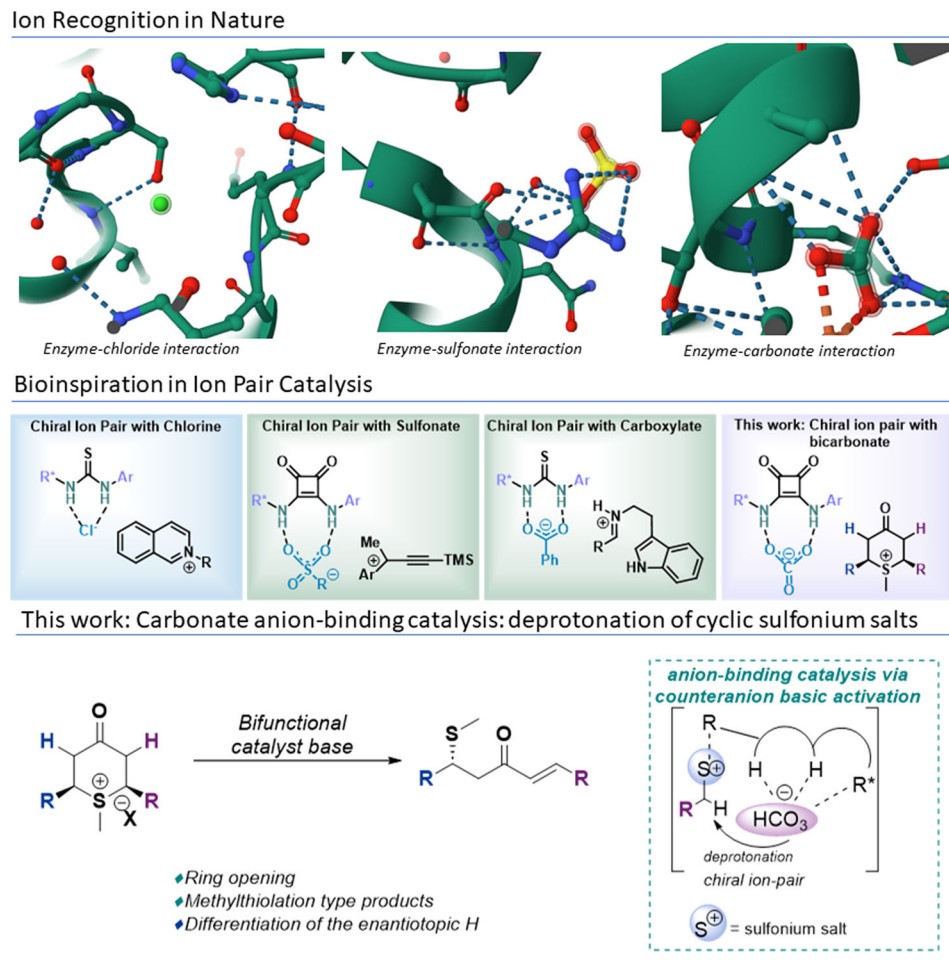

**Fig. 1 | Previous work.** Top: Ion recognition of chlorine, sulfonate, and carbonate anions by enzymatic systems. Middle: anion recognition in asymmetric anion-binding catalysis. Bottom: This work. Organocatalyzed deprotonation of sulfonium salts by anion-bicarbonate-binding catalysis.

highly enantioselective reactions[8,9,12,13,20,21]. In this sense, it should be also highlighted that carboxylate anions have been successfully activated by HBD core-based oraganocatalysts[9,22,23]. However, although more than 500 enzymatic/carbonate anion structures[14] have been described (top-right, Fig. 1)[24], to the best of our knowledge, bicarbonate anion has never been recognized/activate by HBDs-based organocatalysts for an anion-binding process in enantioselectivity-determining processes (For a selected example of bicarbonate anion recognition by HB-donors, see, refs. 25,26). On the other hand, with the growth of organocatalysis, many catalytic desymmetrization methodologies have been developed in the last years[27–32]. However, only few organocatalytic asymmetric deprotonations have been described[33–39]. In this sense, a direct deprotonation by the catalyst, that acts as a Brønsted base[33–36], or the use of a co-base may be implied[37–39]. Thus, the organocatalytic desymmetrization of *meso* epoxides[33] and aziridines[34] has been achieved by introducing a ketone moiety in order to activate the α-protons making them acid enough to be deprotonated by the Brønsted base moiety (tertiary amine). Contrary to what has been previously reported in anion-binding catalysis, in which the catalyst allowed the appropriate approximation of the negatively charged nucleophile (Fig. 1-right and bottom)[9], we envisioned that a squaramide-based organocatalyst could be capable of activating/recognizing an anionic Brønsted base (bicarbonate anion) forming a host-guest complex with the ability to differentiate between two enantiotopic protons (bottom, Fig. 1). To the best of our knowledge, this desymmetrization process would take place by means of an alternative and innovative anion-binding approach, where an asymmetric deprotonation is promoted by the HCO$_3^-$ counteranion recognition of the squaramide organocatalyst, representing a new enantioselective deprotonation strategy (bottom, Fig. 1). Therefore, taking advantage of the high tendency of sulfonium salts to act as leaving groups[40], we envisaged that *meso* sulfonium salt would be an ideal substrate for the asymmetric desymmetrization process via enantioselective deprotonation/ring opening sequence. Moreover, this desymmetrization process represents a ring opening event of a cyclic ketone core in contrast with all previous organocatalytic strategies that maintain this structural motif after the opening of an exocyclic heterocycle such as an epoxide or aziridine. Then, this unconventional strategy would offer the possibility of obtaining new and versatile polyfunctionalized compounds. In fact, this enantioselective deprotonation/ring opening sequence would lead to sulfa-Michael type addition products to ketones. Methodologies for the organocatalyzed enantioselective sulfa-Michael addition using alkyl thiols remains challenging, due to their high pKa[41–44]. Despite the important biological activities of compounds bearing the –SCH$_3$ group and its high synthetic interest, the asymmetric methylsulfenylation remains elusive as the straight addition of methanethiol, which is extremely flammable and toxic compound, has for long hampered its application[41–44].

## Results

Based on our experience[45–49], we began the investigation with the desymmetrization of **1a** as model substrate in the presence of different bases using dichloromethane (CH$_2$Cl$_2$) as solvent. As expected,

**Table 1 | Optimization of the reaction conditions[a]**

| | Catalyst | Solvent | T (°C) | Yield (%)[b] | e.r. (R:S)[c] |
|---|---|---|---|---|---|
| 1 | – | CH$_2$Cl$_2$ | r.t. | n.r. | – |
| 2 | **2a** | CH$_2$Cl$_2$ | r.t. | 87 | 16:84 |
| 3 | **2b** | CH$_2$Cl$_2$ | r.t. | 92 | 35:65 |
| 4 | **2c** | CH$_2$Cl$_2$ | r.t. | 90 | 90:10 |
| 5 | **2d** | CH$_2$Cl$_2$ | r.t. | 88 | 91.4:8.6 |
| 6 | **2e** | CH$_2$Cl$_2$ | r.t. | 76 | 90.3:9.7 |
| 7 | **2f** | CH$_2$Cl$_2$ | r.t. | 84 | 35.7:64.3 |
| 8[d] | **2d** | CH$_2$Cl$_2$ | r.t. | 86 | 87.5:12.5 |
| 9 | **2d** | THF | r.t. | 87 | 94.4:5.6 |
| 10 | **2d** | Toluene | r.t. | 79 | 94:6 |
| 11 | **2d** | C$_6$F$_6$ | r.t. | 82 | 89:11 |
| 12 | **2d** | CHCl$_3$ | r.t. | 90 | 95:5 |
| 13 | **2d** | CHCl$_3$ | 40 | 96 | 70.6:29.4 |
| 14 | **2d** | CHCl$_3$ | 0 | 77 | 96:4 |
| 15 | **2d** | CHCl$_3$ | −20 | 65 | 92.7:7.3 |

[a]Standard reaction conditions: 0.05 mmol of **1a**, 0.06 mmol of NaHCO$_3$ and 0.01 mmol of **2** (20 mol%) in 0.25 mL of solvent.
[b]Isolated yield.
[c]Enantiomeric ratio was determined by supercritical fluid chromatography (SFC).
[d]10 mol% of catalyst was used.

stronger bases were able to afford the elimination product **3a** with high yields (see Supplementary Information). However, when a weaker base such as NaHCO$_3$ was used, there was no observation of **3a** and only **1a** was recovered from the crude (Table 1, entry 1). Therefore, we decided to use NaHCO$_3$ as a base to ensure that no background reaction was taking place. Then, the reaction was carried out in the presence of different organocatalysts with a hydrogen bond donor core bearing a tertiary amine in their structure (Table 1, entries 2–7 and Supplementary Information). To our delight, all catalysts led to the formation of **3a** in good yield as single product. Among all of them, squaramide-based catalysts gave generally very good results in terms of yield and enantioselectivity (Table 1, entries 2–5), achieving the best results when cinchonine-squaramide organocatalyst **2d** was used (Table 1, entry 5). Cinchonine-thiourea organocatalyst **2e** barely affected the enantioselectivity while the yield notably decreased (Table 1, entry 6). Interestingly, when the natural product cinchonine **2f** was used, a significant depletion in the enantioselectivity was observed (Table 1, entry 7), suggesting that the presence of the squaramide or thiourea unit might be relevant in the enantioinduction of the process. Lowering catalyst loading of **2d** to 10 mol%, decreased

the enantioselectivity (Table 1, entry 8). Subsequently, different solvents were studied (Table 1, entries 9–12). Chloroform resulted to be the optimal solvent as both the enantioselectivity and reactivity were increased (Table 1, entry 12). Then, the effect of the temperature was evaluated (Table 1, entries 13–15 and Supplementary Information). As expected, a higher temperature of 40 °C led to an increase yield in detriment of the enantioselectivity (Table 1, entry 13). On the other hand, lowering the temperature to 0 °C diminished the yield of **3a**. When the reaction was carried out at −20 °C, the enantioselectivity and yield of the process were decreased (Table 1, entry 15).

With the optimized conditions in hand (Table 1, entry 12), we studied the scope of the reaction considering the stereoelectronic nature of the aromatic rings (Fig. 2). Electron-withdrawing groups at *para* position of the aromatic rings were very well tolerated, leading to the desired opened methylsufenylated product with excellent yields and enantioselectivities (**3b** and **3c**). Cyclic sulfonium salt-bearing aromatic rings decorated with different halogens in *para* position were also studied (**3d–f**). All *p*-Cl (**3e**), *p*-Br (**3f**), and *p*-F (**3d**) were efficiently desymmetrized with excellent yields and very good enantioselectivities. The introduction of a strong electron-donating substituent by

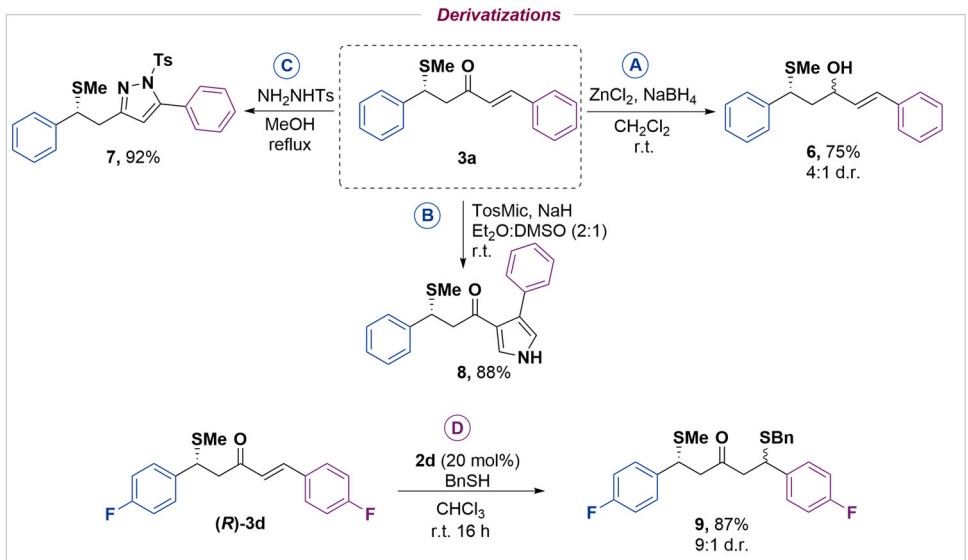

**Fig. 2 | Survey of cyclic keto sulfonium salts.** Reactions were performed on a 0.05 mmol scale of **1**, 0.06 mmol of NaHCO₃ and 0.01 mmol of **2d** in CHCl₃ (0.25 mL) r.t. Isolated yields are shown. Enantiomeric ratios were determined by SFC. [a]0.2 mmol scale of **1**. [b]1 mmol scale of **1**. [c]A mixture 1:1 of CHCl₃ (0.25 mL) and water (0.25 mL) at 0 °C was used.

**Fig. 3 | Derivatizations of 3.** Carbonyl reduction (**A**), heterocycle synthesis from the alkene moiety (**B** and **C**) and a diastereoselective sulfa-Michael addition (**D**).

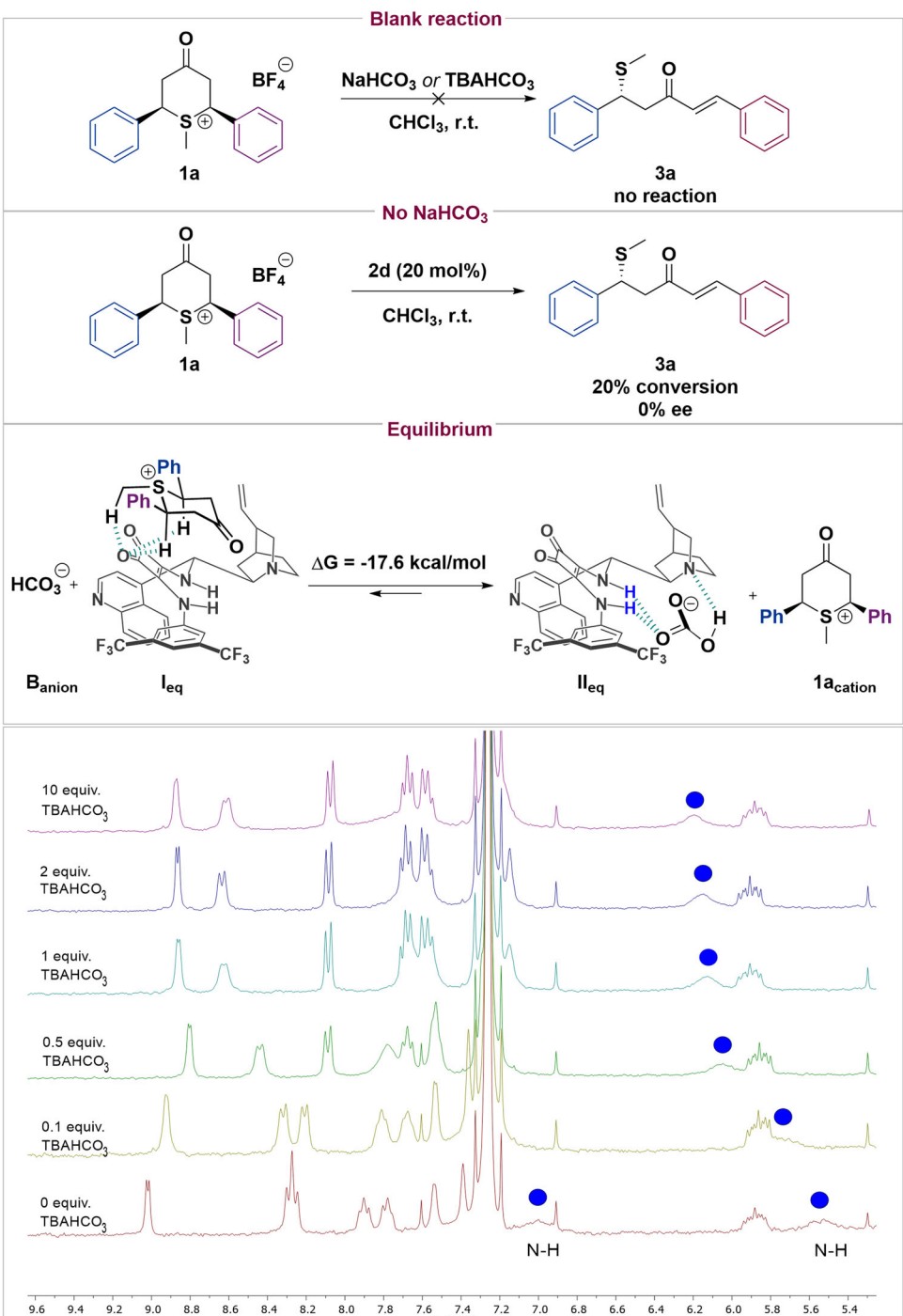

**Fig. 4 | Mechanistic studies.** Control experiments and equilibrium between **1a_cat** and **B_anion** with catalyst **2d**. ¹H-NMR Titration of catalyst **2a** with TBAHCO₃.

mesomeric effects on the aromatic ring such as *p*-MeO effectively afforded **3g** under slightly modified reaction conditions. The use of a *p*-Me substituent (**3h**) presented similar results in terms of yield and enantioselectivity when compared with **3g**. The incorporation of substituents in other positions of the aromatic ring (**3i**, **3j**) was also studied. Although *meta* substitution afforded **3i** in high yield and very good enantioselectivity, *ortho* substitution led to the formation of **3j** in a racemic manner, what could be a consequence of the higher steric effects presented in the substrate. We then wondered if our reaction conditions could tolerate the presence of larger aromatic systems (**3k**, **3l**). In this sense, the introduction of 2-naphtyl could afford **3k** in high yield and slightly lower enantioselectivity, while the incorporation of a

biphenyl system performed the reaction efficiently and **3l** was obtained in excellent yield and very good enantioselectivity. To our delight, the reaction proceeded efficiently when performing the reaction at 0.2 and 1 mmol scale (20 times scale-up) of **1a**.

In the pursuit of extending this methodology to different substrates, we were interested in expanding the desymmetrization process to other sulfur containing heterocycles. To our delight, the corresponding sulfonium salt of 8-thiabicyclo[3.2.1]octan-3-one (**4**) provided promising results in the desymmetrization of other sulfur containing heterocycles, affording **5a** with excellent yield and moderate enantioselectivity. In addition, the ethanethiol derivative **5b** was obtained with a similar efficiency. The absolute configuration of the

stereocenter was unequivocally assigned as (R) using X-ray crystallographic analysis of product **3a** (see Supplementary Information) (CCDC 2160424 (**3a**). The crystallographic data can be obtained free of charge from The Cambridge Crystallographic Data Centre). The same stereochemical outcome was assumed for all the compounds in the scope.

Attracted by the potential functionalization of the final products, we investigated the synthetic value of the obtained methylsulfenylated products. At first, a stereoselective reduction of the carbonyl group was performed using a mixture of $ZnCl_2$ and $NaBH_4$, affording alcohol **6** with good diastereoselectivity and preserving the enantioselectivity of the starting material (Fig. 3A, see Supplementary Information). We then took advantage of the α,β-conjugated system. In addition, and to further exploit the high functionality and versatility of the obtained products, we synthesized interesting imidazole (**7**) and pyrrole (**8**) derivatives in high yield in a single step and without erosion of the presettled stereocenter (Fig. 3B, C, see Supplementary Information).

In addition and to our delight, by using the same catalytic system than the one used for the desymmetrization reaction, we were able to perform a thiolation using benzyl thiol with very good diastereoselectivity and without affecting the enantioselectivity of the starting material (Fig. 3D, see Supplementary Information). Therefore, interesting double conjugated enantioselective thiolated type product can be achieved sequentially with the same catalytic system.

## Mechanism proposal

Aiming to further understand the origins of the enantioinduction and the mechanism in the desymmetrization process, we carried out different experiments and density functional theory (DFT) calculations at the M06-2X/6-31 G(d,p) level[50], followed by single point calculations at the M06-2X/6-31 + G(d,p), we have included solvent effects (chloroform) using the implicit PCM model[51] both in the geometry optimization and in the single point energy calculations. The reaction takes place through an enantioselective deprotonation of one of the methylenes of **1a** followed by elimination of the sulfonium functionality. This deprotonation step can be performed by either the quinuclidine moiety of catalyst **2d** or by the $NaHCO_3$ present in the solution. Quinuclidine might be considered the base that perform the deprotonation as $HCO_3^-$ ($NaHCO_3$ or $TBAHCO_3$) has been demonstrated to be not basic enough to deprotonate **1a** (blank reaction Fig. 4). In order to understand the role of $NaHCO_3$, we perform a reaction in absence of it. The reaction proceeded with the same conversion as the catalyst loading (20 mol%), which indicates that the catalyst **2d** is unable to be regenerated and only capable of performing one catalytic cycle (see Fig. 4). In addition, and surprisingly, the enantioselectivity was completely inhibited. This fact suggests that catalyst **2d** is basic enough to deprotonate **1a**, but it cannot be the chiral catalytic species that differentiates between the two enantiotopic protons during the asymmetric deprotonation. Therefore, this indicates that $NaHCO_3$ is involved in the enantiodetermining step and that its role is not only to regenerate the catalyst. In light of this experimental evidence, we studied the equilibrium shown in Fig. 4 by DFT calculations. Thereby, catalyst **2d** was found to preferably binds $B_{anion}$ than $1a_{cation}$ by 17.6 kcal/mol. This result indicates that even though species $I_{eq}$ could lead to the direct deprotonation by the quinuclidine moiety of catalyst **2d**, its formation is avoided when $HCO_3^-$ is present in the media. Hence, catalyst **2d** mainly forms species $II_{eq}$, which is in fact activating the $HCO_3^-$ and enhancing its basicity. In addition, we performed a NMR

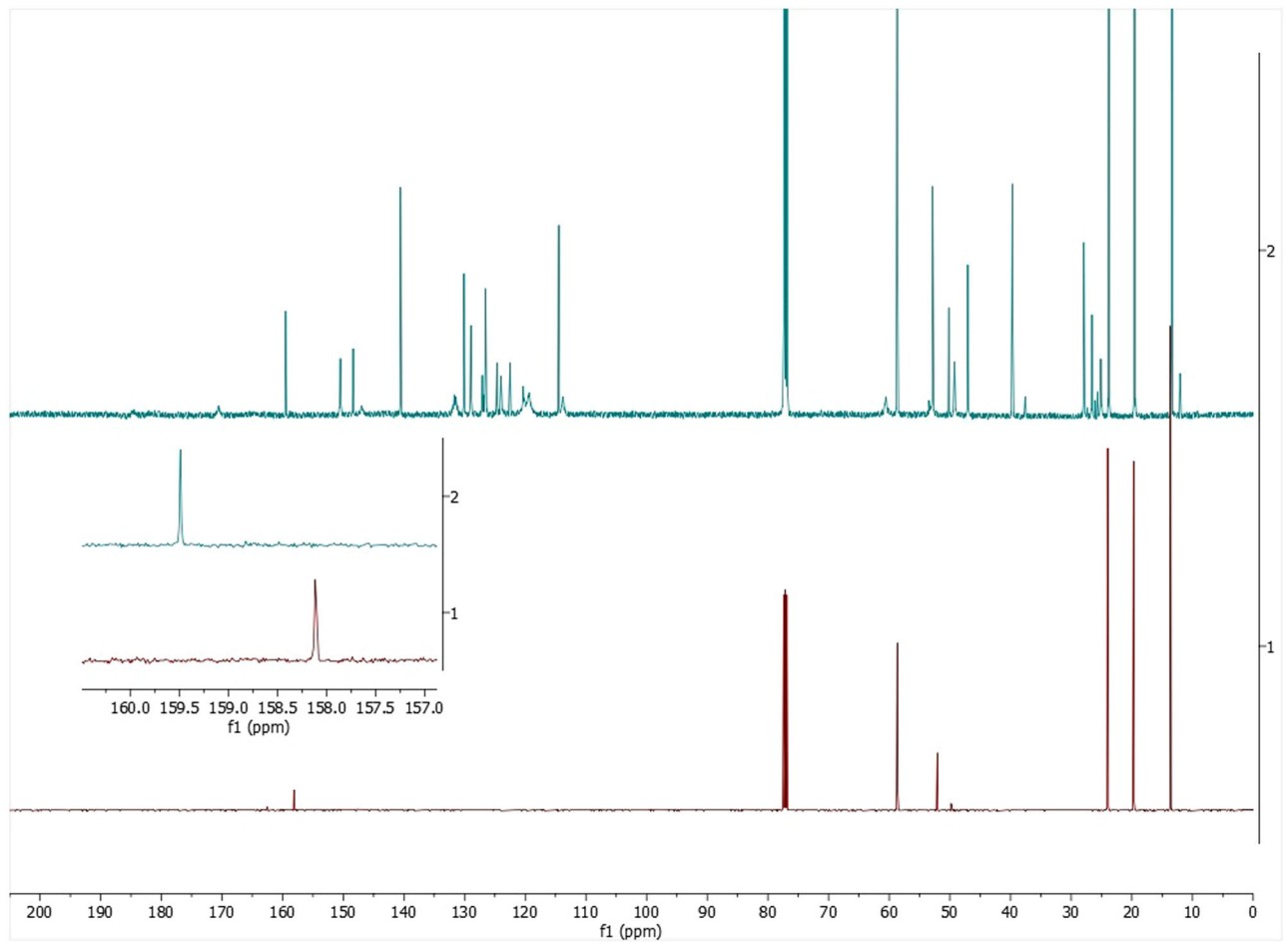

**Fig. 5 | $^{13}$C NMR study.** $^{13}$C NMR Spectra of a mixture 1:1 of the catalyst **2d** and TBAHCO$_3$ (tetrabutylammonium bicarbonate) in CDCl$_3$ (final amount of 0.6 mL).

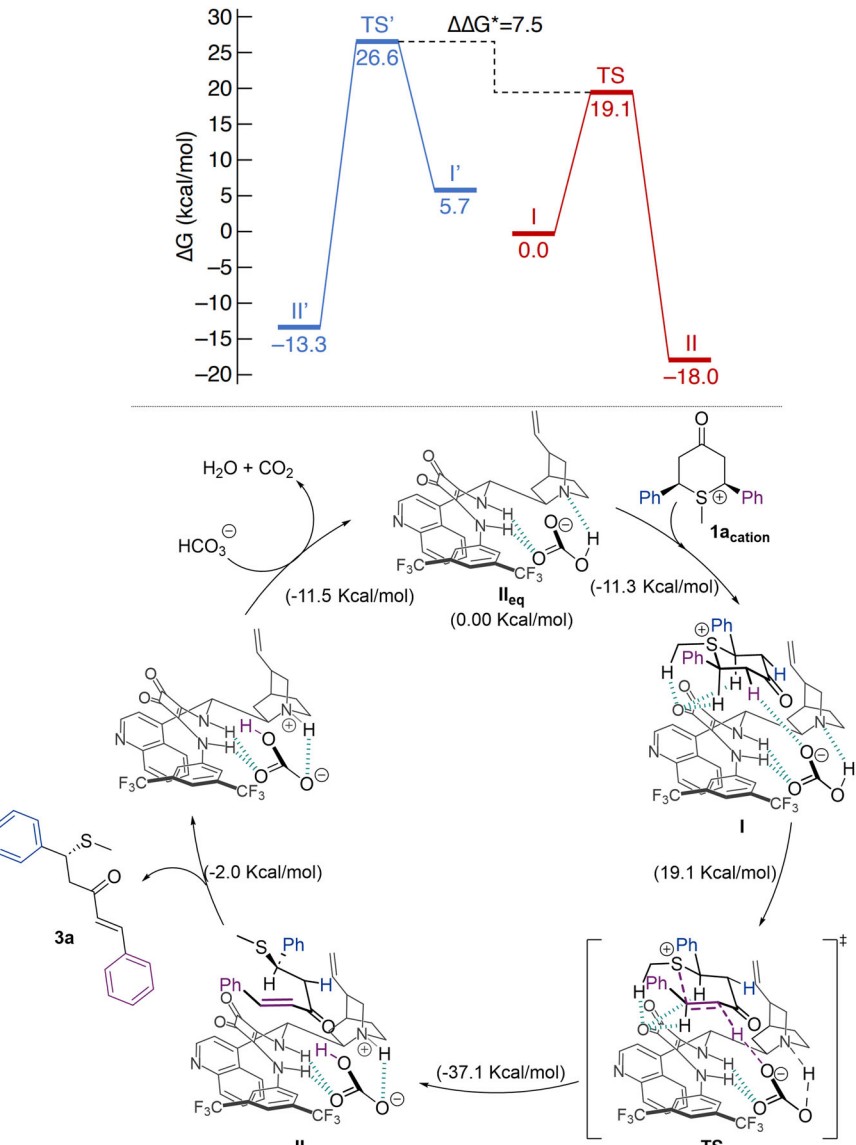

**Fig. 6 | Proposed reaction mechanism and confirmatory studies.** Top: Energy profile. DFT M06-2X/6-31 + G(d,p)//M06-2X/6-31 G(d,p) (PCM = chloroform) reaction energy profile for both enantiomer (major in purple, minor in blue). Bottom: proposed catalytic cycle.

titration of catalyst **2d** (see Supplementary Information) with TBAHCO$_3$ (tetrabutylammonium bicarbonate), and the proton signals of the N–H bonds of the squaramide are clearly shifted downfield after the addition of only 0.1 equiv. of TBAHCO$_3$. There are also changes in the chemical shifts of aromatic H signals.

In addition, a similar effect was observed by $^{13}$C-NMR of catalyst **2d** with TBAHCO$_3$ (see Fig. 5). The carbon signal of the HCO$_3^-$ was shifted downfield in the presence of 1.0 equiv. of TBAHCO$_3$. Moreover, the NMR studies showed a great solubility effect (see photos in Supplementary Information) that might emphasize the interaction between the catalyst and the HCO$_3^-$. Catalyst **2d** is totally insoluble in CDCl$_3$, by contrast, it gets solubilized immediately when mixed with the base. All this support that bicarbonate anion is recognized by the squaramide core, forming a host-guest complex and promoting the proposed anion-binding catalysis.

Once **II$_{eq}$** is formed, **1a$_{cation}$** can coordinate to catalyst **2d** by hydrogen bonding and forms the chiral hydrogen-bonded complex **I** as illustrated in Fig. 6. Supported by literature, α-protons to the sulfur atom in **1a$_{cation}$** might be considered acid enough to act as hydrogen bond donors[52,53]. Thus, coordination to the carbonyl of the squaramide

moiety is feasible. Notice that the sulfonium cation and the bicarbonate anion are not directly interacting in complex **I**, but they are both bonded through intermolecular non-covalent interactions to the catalyst. Mainly hydrogen bonds stabilize the complex and it can thus be seen as a hydrogen-bond-assisted ion-pair[54].

To further prove the coordination of the bicarbonate and substrate reactants with the squaramide catalyst, we carried out an exhaustive exploration of the conformational space; several structures were sampled by performing molecular metadynamics with the GFN2-xTB Hamiltonian[55,56] implemented in the CREST tool[57], including solvent effects within the analytical linearized Poisson–Boltzmann (ALPB) model[58]. In these simulations, the algorithm was set to search for the most stable conformers within an energy window of 15 kcal·mol$^{-1}$, obtaining a total of 3041 structures, with different relative orientations of the bicarbonate and substrate with respect to the squaramide. This study followed a computational strategy recently implemented by Trujillo and coworkers[59], where a complete exploration of the conformational space of different phase-transfer catalysts was presented. The results of our exploration are summarized in Fig. 7. Analysis of the coordination of the bicarbonate and substrate reactants with the

squaramide catalyst over 3041 structures found in the chemical space exploration, was performed at the GFN2-xTB level of theory. $R_1$ is the distance between bicarbonate and squaramide, measured as the smallest distance between one carboxylate oxygen atom in the bicarbonate and one hydrogen atom in the NH of the squaramide. $R_2$ is the substrate squaramide distance, measured as the smallest distance between the carboxylic oxygen atom in the substrate and one hydrogen atom in the NH of the squaramide.

The analysis of the distances shows that most of the structures present a direct coordination between the bicarbonate and the squaramide through hydrogen bonds ($R_1 \sim 1.7$ Å). Indeed, among all the structures, only 48 show $R_1 < 2$ Å, i.e., 1.6% in the energy window considered, being the lowest in energy at 5.0 kcal·mol⁻¹ with respect to the most stable one. Exchange of positions between the bicarbonate and the substrate, i.e., $R_1 > 2$ Å and $R_2 < 2$ Å, is only found in two structures, being the most stable at 7.9 kcal·mol⁻¹ with respect to the most stable one. These facts show the great tendency of the organocatalyst to activate the bicarbonate anion forming a complex with sulfonium species. An exhaustive analysis to characterize bonding properties in this complex has been carried out using different computational techniques (EDA-NCI[60,61], NCIPLOT[62–64], QTAIM[65,66], see Supplementary Information). They have shown that the main stabilizing interaction is electrostatic, originated by the positive charge of the sulfonium cation and the bicarbonate, although other non-covalent interactions (NCIs), such as hydrogen bonds or π-π stacking, play also a non-negligible role.

After formation of **I**, the deprotonation step takes place with an energy barrier of 17.1 kcal/mol. Notice that the relative orientation of the sulfonium in structure **I** (Fig. 6) is different to that shown in Fig. 7c, favoring its deprotonation (see details in the Supplementary Information). Final product **II** is directly formed from this deprotonation step in an exergonic process (−20.0 kcal/mol). An analysis of the geometry of the transition state (**TS**) for this process reveals a simultaneous elongation of $C_{benzylic}$−S bond (from 1.86 to 1.90 Å) and $C_{methylenic}$−H bond (from 1.10 Å to 1.45 Å) and a shortening in $C_{methylenic}$-$C_{benzylic}$ bond (from 1.53 to 1.49 Å). The O−H distance corresponding to the proton abstraction in the TS is 1.174 Å, characteristic of a covalent bond (see Fig. 8), also confirmed with the presence of a bond critical point in the QTAIM simulations (see Supplementary Information). Considering that no other minimum was found in the potential energy surface (PES) from this TS (see Supplementary Information) and the geometrical features of **TS**, the elimination step might be taking place at the same time as the deprotonation one. Thus, the reaction pathway resembles more of an *anti*-E2 mechanism. This is also in agreement with the fact that only *E* configuration is observed for the double bond of **3a**, as a consequence of the *anti*-disposition of the sulfonium leaving group and the H that is deprotonated in the TS. Lastly, it was assumed that the regeneration of the catalytic species **II$_{eq}$** could take place by means of another $HCO_3^-$, producing $CO_2$ and $H_2O$ in the process, being the formation of these compounds the reason why the cycle is thermodynamically favorable. Having described a plausible mechanism for the desymmetrization process, we decided to study the enantioselectivity of the reaction by DFT calculations at the same level. An initial approach **I'** from the other face of catalyst **2d** was found to be optimal as the starting

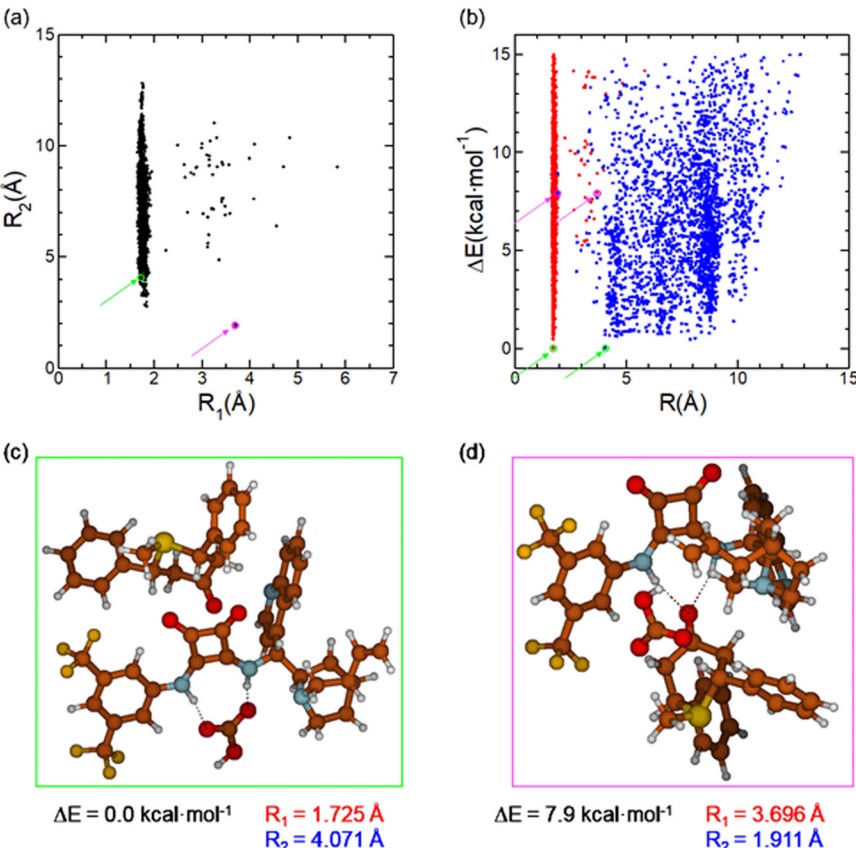

**Fig. 7 | Analysis of the coordination of the bicarbonate and substrate reactants with the squaramide catalyst. a** $R_1$ as a function of $R_2$ for each structure. **b** $R_1$ (in red) and $R_2$ (in blue) as a function of the relative energy between the structures. **c** Most stable structure. **d** Most table structure where the $R_2$ is smaller than $R_1$.

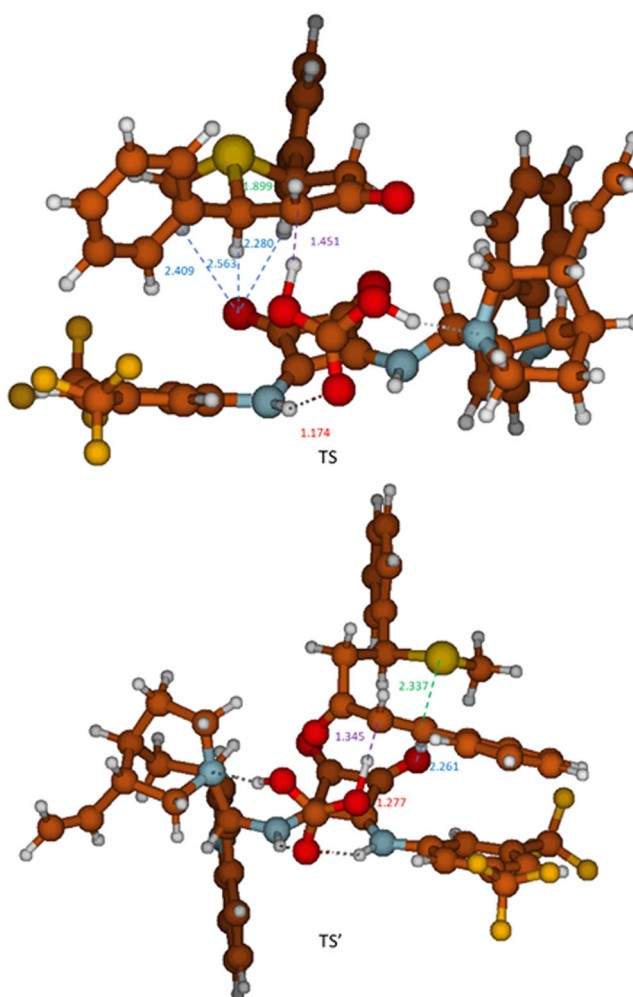

**Fig. 8 | Transition states.** Top: TS for the major enantiomer. bottom: TS' for the minor enantiomer. Distances marked in purple are used for the bonds which are present in the elimination. Distances marked in greenish-blue are for hydrogen bonds.

approximation (Fig. 6 blue line). Again, final product **II'** was obtained in an exergonic process (−19.1 kcal/mol) via an energy barrier of 20.9 kcal/mol, which is higher than the one found for the major enantiomer. A comparison of both TS structures shows significant differences between the geometries and distances (Fig. 8). While for the major enantiomer, **TS** shows a $C_{benzylic}$−S distance of 1.90 Å; for the minor enantiomer, **TS'** presents a $C_{benzylic}$-S distance of 2.34 Å. Therefore, the generation of a higher energetic benzylic carbocation might be taking place, resulting in a higher energetic barrier for the minor enantiomer (see Supplementary Information).

## Discussion

In conclusion, we have reported the enantioselective desymmetrization of cyclic keto sulfonium salts through enantioselective deprotonation. The presented anion-binding catalyzed enantioselective desymmetrization takes place through recognition of $HCO_3^-$ anion by the squaramide organocatalyst forming an active catalytic species able to differentiate between the two enantiotopic protons, triggering the ring opening event taking advantage of the great tendency of sulfonium salts to act as leaving groups. The ring opening process gives access to β-methylsulfenylated sulfa-Michael addition type products with excellent yields and

very good enantioselectivities. The bifunctional organocatalyst has been demonstrated to be capable of activating simultaneously the base and the keto sulfonium salt by DFT calculations and experimental evidences.

## Methods

In a typical enantioselective desymmetrization, the corresponding sulfonium salt (**1**) (1 equiv., 0.05 mmol), sodium bicarbonate (1.2 equiv., 0.06 mmol, 5 mg), and **2d** (0.2 equiv., 0.01 mmol, 6 mg) were dissolved in chloroform (0.25 mL) and stir at room temperature overnight. Then, diethyl ether was added (2 mL) and the reaction crude was filtered. The filtrate was concentrated under reduced pressure and purified by flash chromatography.

## Data availability

All data supporting the findings of this study, including experimental details, spectroscopic characterization data for all compounds are available within the article and the Supplementary Information section, or from the corresponding author upon request. Crystallographic data for the structure reported in this Article have been deposited at the Cambridge Crystallographic Data Centre, under deposition numbers CCDC 2160424 (**3a**). Copies of the data can be obtained free of charge via https://www.ccdc. cam.ac.uk/structures/. Source data are provided with this paper.

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

## Acknowledgements

Financial support was provided by the Spanish Government (PID2021-122299NB-I00, TED2021-130470B-I00, TED2021-129999B-C32, PID2022-138470NB-I00), 'Comunidad de Madrid', European Structural Funds (S2018/NMT-4367), proyectos sinérgicos I + D (Y2020/NMT6469) and Comunidad Autónoma de Madrid (SI1/PJI/2019–00237). J.A.F.-S. thanks the Spanish Government for a Ramón y Cajal contract. We acknowledge the generous allocation of computing time at the Centro de Computación Científica of the Universidad Autónoma de Madrid (CCC-UAM).

## Author contributions

J.H.-M., R.R., J.A.F.-S., and J.A. designed the experiments and analyzed the data. J.H.-M. and R.R. performed the experiments. F.A.-G. and S.D.-T. performed and analyzed the quantum chemistry calculations. R.R.-R. and J.A.F.-S. described the Supplementary Information. The manuscript was written through contributions of all authors. All authors have given approval to the final version of the manuscript.

## Competing interests

The authors declare no competing interests.
