## [Peer Review File · Nature Communications]

Bicarbonate-Binding Catalysis for the Enantioselective Desymmetrization of Ketosulfonium saltsReviewers' Comments:

Reviewer #1:

Remarks to the Author:

Fernandez-Salas and co-workers reported on a desymmetrization of keto sulfonium salts by embracing the binding and activation of bicarbonate anion by a hydrogen-bond-donor organocatalyst, thus enabling an enantioselective deprotonation/ring opening reaction to form highly valuable β -methylsulfenylated synthetic building blocks. Computational studies, combined with titrations and further experimental evidences, supported the proposed key bicarbonate anion-binding by the catalyst. This work represents the first example in which the binding of bicarbonate anion has been recognized as key event for allowing high levels of enantioinduction in an organocatalytic process, which might inspire and open new possibilities in the design of further asymmetric transformations with this type of normally considered weak binding anion. Therefore, this work might be suitable for publication in the journal Nat. Commun. after major revision upon addressing the following:

- 1) Title: the title should be a bit more precise and include important information about the work is then presented, such as "enantioselective desymmetrization of cyclic ketosulfonium salts"
- 2) Page 2, introduction: This reviewer disagrees with the statement of the authors: "Following the path created by Jacobsen's pioneer studies on the recognition of nitrile anions by urea and thiourea based chiral organocatalysts, 10 other research groups have described different catalyst designs." In this early work of Sigman and Jacobsen on the Strecker reaction, the possibility of anion recognition was not considered, for which it cannot be taken as first or pioneer example being the reference of anion-binding organocatalysis. Please correct this and provide a more accurate example of first or pioneer work in anion-recognition in asymmetric organocatalysis
- 3) Scheme 1: the authors should also include in the scheme a figure with the more related recognition/binding of carboxylates from the literature, rather than only halogen and sulfonate anions. Moreover, in the Scheme 1, Middle. The example of ion pair with chloride is not correct. When using thioureas as organocatalysts, the substrate that should be displaced is an isoquinoline, not quinoline. Also, the original work of Jacobsen represented here with this figure should be cited.
- 4) Scheme 1: please provide a short information in the figure of Scheme 1 top on what it is to seen for each case below the cartoon, e.g. enzyme-chloride interaction or chloride-binding protein, etc....
- 5) Page 4: "Jacobsen's group has recently reported a squaramide based bifunctional organocatalyst system that activates the sulfonate anion. 11,16,17" reference 11 is not from Jacobsen. Please remove or provide the right literature at this position.
- 6) Page 4: The authors claimed that the bicarbonate anion has never been recognized by HBD organocatalysts. Although this type of binding/recognition has been previously described, it is true that has not been exploited/considered in catalysis before. However, it would be desirable to add some references for the bicarbonate anion recognition by HB-donors
- 7) Table 2, scope: Is this method limited to diaryl keto sulfonium salts 1? Except for the bicyclic derivatives 4, it is possible to enrol aryl-alkyl or dialkyl substituted substrates? Please provide a few examples.
- 8) Table 2: the scale used in this study is rather small, 0.05 mmol, and the yields given then questionable. Please provide some upscaling reactions to at least 0.2 mmol and one gram scale reaction.
- 9) Table 1 and Scheme 2: a reaction with a substrate bearing a complex moiety, and/or the derivatization of the products to a bioactive compound should be provided to show potential applications of this method
- 10) Scheme 3: please include as blank also the reaction with TBAHCO₃ as phase transfer (PT)/more soluble salt in the organic solvent used. Is then the product formed? If this is the case, the organocatalyst can be considered as PTC
- 11) Scheme 3, NMR titration: please indicate the ortho-protons of the 3,5-(CF₃)₂Ph group that are also participating in the binding and are clearly shifted
Additionally, other peaks corresponding to the quinoline core suffer a shift in the ¹H NMR. Which H are

those and why it is the interaction leading to this observation? Could you comment on that? Could some information in this regard be gained by re-analysis of the structures and transition states from the calculations?

12) Page 12: "In addition, a similar effect was observed by ^{13}C -NMR by titration of catalyst 2d with TBAHCO_3 (see Figure 1)." Figure 1 cannot be considered a titration since only one experiment with 1 equiv. of TBAHCO_3 is shown. Please correct this statement.

13) Reference 8: the name of one of the authors is not given properly.

14) Supporting Information:

- This version of the SI seems not to be the final one, since some comments from the authors are still included. See e.g. page S46.

Please provide a final, comment-free version.

- Page S5: in schemes of Procedure C and D the equivalents of the base is shifted up on the reaction arrow

- Page S5, Procedure E: It should be "Sulfonium salt 4"

- Pages S22-23, titration: The NMR set in page S22 is rather small and it is difficult to see. Please provide a bigger one

In the expanded NMR titration set in page 23, D_2O is not well displayed.

- Page S23: how was the binding constant determined, using only one NH of the squaramide catalysts as shown in the Titration Figure at the end of the page? Which model and stoichiometry of the Catalyst-Bicarbonate anion complex have been used for the fitting? It would be desirable to include further H points for adjusting in BindFit. Please also provide the fitting from BindFit.

- For a few spectra the vertical axis label is given. For consistency, please remove this for these cases (e.g. 5b, etc)

- Moreover, for the ^1H NMR of 5b a box on the signals at 3.0 and 1.5 ppm can be noticed when going with the mouse over it.

Same strange boxes can also be notices in other spectra.

Please confirm that these NMRs have not been manipulated.

- The ^{13}C NMR of some products is rather diluted (e.g. 3b, etc). Please provide a better NMR for those cases.

- Page S59, the racemic chromatogram is cut. Please provide the same range from 0 min as for the product from the catalytic reaction

15) In general, several typos could be find along the manuscript and SI. Please check carefully and correct when necessary.

Reviewer #2:

Remarks to the Author:

Overall, this is a nice paper, but unfortunately, the reviewer is not convinced that the work is of wide enough impact to merit publication in Nature Communications.

The authors claimed: "Anion-binding processes are well-known for their crucial role in molecular recognition. This type of phenomenon has only recently been utilized for catalysis." This kind of interaction has been used in ion-pair catalysis. Therefore, although it is new for desymmetrization, it is just one example, not an extensive scope. Can it be transferred to other symmetrical species? Can it be done with other anions?

In Scheme 1, the authors introduce the concept of a "chiral close ion-pair." However, the description of this interaction as a "close ion-pair" may not be entirely accurate. The positioning of charges and the distance between them raise concerns, as there is a complex web of interactions, including hydrogen bonds, dipole-dipole forces, and various non-covalent interactions (NCIs), which complicates the oversimplified label of "ion-pair." Additionally, the anion of the sulfonium salt in this model is not clearly defined, and its role in the computational study remains unclear.

The approach adopted in this study follows a somewhat traditional sequence. A more contemporary approach would involve starting with a computational study to gain a thorough understanding of the mechanism, followed by experimental validation, and then proposing further computational investigations for subsequent validation.

One notable omission is whether the solvent was considered in the optimization process, which can significantly impact reaction pathways and product outcomes. It is not clear from "including solvent effects (chloroform) using the implicit PCM model in all the cases", including in the optimizations or in the SP calculations?

Examining Figure 2, it is evident that in the left panel, where $DE=0.0$ kcal/mol, the charged species are notably distant from each other. This observation raises doubts about whether the described "close ion-pair" is an accurate characterization. The assertion that it represents a pure ion-pair structure appears ambitious and speculative.

Comparing structure I from Scheme 4 with the most stable conformer according to CREST in Figure 2, it becomes apparent that these are not identical structures. The 3D representation in Figure 2 shows the carbonate forming a parallel hydrogen bond with the squaramide, while the 2D structure in Scheme 2 exhibits a bifurcated bond. This discrepancy prompts questions regarding the mechanism of deprotonation, especially considering the distance between the sulfonium salt and the bicarbonate. A more comprehensive explanation of the pre-transition state assembly is required, as well as clarification on how to transition from the most stable conformer to TSI as presented in Table S3. This transition involves substantial reorganization of hydrogen bonds, and the change in the deprotonation site remains unexplained.

Furthermore, the notably short H-O bond lengths within the transition states seem to suggest that the bond is already formed, which warrants further discussion and clarification.

The authors claim that the mechanism for the minor enantiomer resembles an E1 mechanism, unlike the major enantiomer, which shows similarities to an E2 mechanism. However, the absence of a minimum corresponding to the carbocation raises doubts. It would be prudent to perform an intrinsic reaction coordinate (IRC) analysis to confirm the reaction pathway, which should provide straightforward validation.

Regarding TS', is there any steric issue between the phenyl group of the sulfonium and the squaramide catalyst?

Given that this work is based on hydrogen bonds and weak interactions, a more comprehensive analysis and characterization of non-covalent interactions (NCIs) upon complexation are crucial to elucidate the underlying rationale for the study's findings. Additionally, if the authors wish to discuss ion-pair interactions, an analysis of the nature of these interactions through tools like Quantum Theory of Atoms in Molecules (QTAIM) and energy decomposition analysis is necessary to substantiate their claims.

In Table 1, which discusses the catalyst scope, it is evident that replacing OMe with H has a substantial impact. It is essential to consider whether a computational study was conducted to elucidate the reasons behind this effect. Moreover, the choice of squaramide over thiourea in cat 2e, which yielded the best experimental outcomes, should be explained.

In summary, this study raises numerous questions, making it an interesting yet incomplete piece of work. The claim of groundbreaking anion recognition may require further substantiation, and while it shows promise, it may not meet the criteria for publication in a prestigious journal like Nature.

why? Furthermore, cat 2e shows the best experimental outcomes with room temperature, a 90% yield, and a 95:5 enantiomeric ratio. Why choose squaramide over thiourea?

There are many questions that make this work interesting but incomplete. Additionally, the idea of anion recognition may not be as groundbreaking as suggested in the introduction. This is a very promising piece of work that could be published in a high-impact factor journal, but perhaps not in Nature.

Comments referee 1:

Fernandez-Salas and co-workers reported on a desymmetrization of keto sulfonium salts by embracing the binding and activation of bicarbonate anion by a hydrogen-bond-donor organocatalyst, thus enabling an enantioselective deprotonation/ring opening reaction to form highly valuable β -methylsulfenylated synthetic building blocks. Computational studies, combined with titrations and further experimental evidences, supported the proposed key bicarbonate anion-binding by the catalyst. This work represents the first example in which the binding of bicarbonate anion has been recognized as key event for allowing high levels of enantioinduction in an organocatalytic process, which might inspire and open new possibilities in the design of further asymmetric transformations with this type of normally considered weak binding anion. Therefore, this work might be suitable for publication in the journal Nat. Commun. after major revision upon addressing the following:

We would like to thank the reviewer for his/her careful reading of our manuscript, for his/her appreciation considering it as first example in which the binding of bicarbonate anion in a catalytic process.

1) Title: the title should be a bit more precise and include important information about the work is then presented, such as “enantioselective desymmetrization of cyclic ketosulfonium salts”

Referee is right. The title has been modified accordingly.

2) Page 2, introduction: This reviewer disagrees with the statement of the authors: “Following the path created by Jacobsen’s pioneer studies on the recognition of nitrile anions by urea and thiourea based chiral organocatalysts,¹⁰ other research groups have described different catalyst designs.”

In this early work of Sigman and Jacobsen on the Strecker reaction, the possibility of anion recognition was not considered, for which it cannot be taken as first or pioneer example being the reference of anion-binding organocatalysis. Please correct this and provide a more accurate example of first or pioneer work in anion-recognition in asymmetric organocatalysis

We thank the referee for the suggestion. We have modified the statement of the introduction for: “Following pioneer studies that established that under appropriate reaction conditions, hydrogen bond donor catalysts may bond to the anionic component of an ion pair, thereby generating a new reactive species,^{10,11} several research groups have described different catalyst designs based on this new mode of activation.” The references have been modified accordingly.

3) Scheme 1: the authors should also include in the scheme a figure with the more related recognition/binding of carboxylates from the literature, rather than only halogen and sulfonate anions.

The scheme has been modified. In addition, discussion about binding of carboxylates and the corresponding references have been included in the new version of the manuscript.

Moreover, in the Scheme1, Midd. The example of ion pair with chloride is not correct. When using thioureas as organocatalysts, the substrate that should be displaced is an isoquinoline, not quinoline. Also, the original work of Jacobsen represented here with this figure should be cited.

The example of ion pair with chloride represented in Scheme 1 as well as the corresponding reference have been corrected.

4) Scheme 1: please provide a short information in the figure of Scheme 1 top on what it is to seen for each case bellow the cartoon, e.g. enzyme-chloride interaction or chloride-binding protein, etc.....

A short information for each case has been included in order to clarify Scheme 1.

5) Page 4: “Jacobsen’s group has recently reported a squaramide based bifunctional organocatalyst system that activates the sulfonate anion.11,16,17” reference 11 is not from Jacobsen. Please remove or provide the right literature at this position.

Reference 11 is a review that describes Jacobsen’s works. In the new version of the manuscript, we have removed reference 11, following referees’ suggestion.

6) Page 4: The authors claimed that the bicarbonate anion has never been recognized by HBD organocatalysts. Although this type of binding/recognition has been previously described, it is true that has not been exploited/considered in catalysis before. However, it would be desirable to add some references for the bicarbonate anion recognition by HB-donors

Two new references about bicarbonate anion recognition by HB-donors with different purposes have been included (see references 25 and 26 of the manuscript)

Scope: Is this method limited to diaryl keto sulfonium salts 1? Except for the bicyclic derivatives 4, it is possible to enrol aryl-alkyl or dialkyl substituted substrates? Please provide a few examples.

The suggested aryl-alkyl sulfonium salts have not been considered. The template proposed would not be a symmetric *meso* compound, which is required to be subjected to our desymmetrization process.

As the referee pointed out, in the manuscript, we have presented two examples of dialkyl substituted substrates (**5a** and **5b**). In addition, we have tried to expand the dialkyl substituted templates. To do so we tried to prepare the alkyl substituted analogous of compounds 1. However, we were unable to prepare those sulfonium salts.

8) Table 2: the scale used in this study is rather small, 0.05 mmol, and the yields given then questionable. Please provide some upscaling reactions to at least 0.2 mmol and one gram scale reaction.

We appreciate referees’ observation. We have tested the reaction at the 0.2 and 1 mmol scale. We have observed that although the results can be considered satisfactory, there is a slight erosion of the enantioselectivity when performed already at 1 mmol scale.

The results obtained have been included in the new version of the manuscript.

9) Table 1 and Scheme 2: a reaction with a substrate bearing a complex moiety, and/or the derivatization of the products to a bioactive compound should be provided to show potential applications of this method

Thank you for your insightful question. Firstly, let's recognize the complexity of the task suggested. Although we have not been able to identify a direct and straightforward transformations towards bioactive compounds, we intensely tested the derivatization of our substrates towards recognizable moieties or functionalities in biologically active compounds such as for example lactones and oxetanes. Unfortunately, we have not been able to prepare any of them. In addition, *meso* sulfonium salts were not effectively modified in order to bear more complex substituents. All this has limited the derivatization of the obtained compounds. However, we showed in the manuscript that,

taking advantage of the activated alkene and the carbonyl resulted from the desymmetrization reaction, we were able to perform a direct 1,4 addition of a thiol and the reduction of the carbonyl in a diastereoselective manner. We believe that these transformations may show the potential of the obtained compounds.

Moreover, I would like to highlight that the development of new methodologies that involves for example new modes of activation as the presented one, would not necessarily require to probe their applicability towards the synthesis of recognizable products at an early stage (for similar examples, see: J. Am. Chem. Soc. 2014, 136, 13999; Science, 2017, 358, 761 among many others). As the referee pointed out “*This work represents the first example in which the binding of bicarbonate anion has been recognized as key event for allowing high levels of enantioinduction in an organocatalytic process, which might inspire and open new possibilities in the design of further asymmetric transformations with this type of normally considered weak binding anion*”. We agree with this statement, and we think that these studies can be seen as a starting point for further development that could potentially lead to the synthesis of interesting and recognizable molecules, as the main objective of the presented manuscript was to demonstrate the possibility to activate bicarbonates anions with squaramide-based organocatalysts in an enantioselective anion binding catalytic process.

10) Scheme 3: please include as blank also the reaction with TBAHCO₃ as phase transfer (PT)/more soluble salt in the organic solvent used. Is then the product formed? If this is the case, the organocatalyst can be considered as PTC

The information of the blank reaction in presence of TBAHCO₃ has been added in the manuscript and in scheme 3.

11) Scheme 3, NMR titration: please indicate the ortho-protons of the 3,5-(CF₃)₂Ph group that are also participating in the binding and are clearly shifted. Additionally, other peaks corresponding to the quinoline core suffer a shift in the ¹H NMR. Which H are those and why it is the interaction leading to this observation? Could you comment on that? Could some information in this regard be gained by re-analysis of the structures and transition states from the calculations?

We have not included any conclusions about the ortho-protons of the 3,5-(CF₃)₂Ph as we have not considered or observed any binding interaction of those protons in any of the transition states. In addition, regarding the shown NMR spectra in Scheme 3, the mentioned protons are difficult to be identified through the different ¹H-NMR of the titration.

However, as the referee pointed out, in the case of the quinoline core there is a clear effect in the ¹H-NMR. We have identified that protons at 8.3 ppm are allocated at carbon 5 and 8 of the quinoline core. Those protons are equivalent and therefore appear as a sole triplet for the two protons. However, and in presence of the anion, that signal split in two doublets clearly differentiated. This may support our hypothesis corroborating the fact that the catalyst environment is clearly modified when bonded to the anion. In TS shown in Figure 3, we can see how proton 5 and 8 of the quinoline are not equivalent anymore due to the disposition of the catalyst with respect to the bicarbonate. Figure S7 (supporting information) shows an interaction between hydrogen of C5 of the quinoline and the quinuclidine ring. This interaction together with the rigidity acquired might help to differentiate them. This discussion has been included in the Supporting information (section 8.2).

12) Page 12: “In addition, a similar effect was observed by ^{13}C -NMR by titration of catalyst 2d with TBAHCO₃ (see Figure 1).” Figure 1 cannot be considered a titration since only one experiment with 1 equiv. of TBAHCO₃ is shown. Please correct this statement.

Referee is absolutely right. We have modified the statement in the new version of the manuscript.

13) Reference 8: the name of one of the authors is not given properly.

The name has been corrected.

14) Supporting Information:

- This version of the SI seems not to be the final one, since some comments from the authors are still included. See e.g. page S46.

Please provide a final, comment-free version.

Final version revised without comments has been provided.

- Page S5: in schemes of Procedure C and D the equivalents of the base is shifted up on the reaction arrow

Fixed

- Page S5, Procedure E: It should be “Sulfonium salt 4”

Fixed

- Pages S22-23, titration: The NMR set in page S22 is rather small and it is difficult to see. Please provide a bigger one

A bigger one has been provided.

In the expanded NMR titration set in page 23, D₂O is not well displayed.

- Page S23: how was the binding constant determined, using only one NH of the squaramide catalysts as shown in the Titration Figure at the end of the page? Which model and stoichiometry of the Catalyst-Bicarbonate anion complex have been used for the fitting? It would be desirable to include further H points for adjusting in BindFit. Please also provide the fitting from BindFit.p

Following referees' suggestion, we have expanded that section of the supporting information. We have added all the information required and also the fitting from BindFit.

- For a few spectra the vertical axis label is given. For consistency, please remove this for these cases (e.g. 5b, etc)

Vertical axis has been removed from all the spectra.

- Moreover, for the ^1H NMR of 5b a box on the signals at 3.0 and 1.5 ppm can be noticed when going with the mouse over it.

Same strange boxes can also be notices in other spectra.

Please confirm that these NMRs have not been manipulated.

We have checked the mentioned box and we have not been able to detect it. It might be a visual effect.

This is the ^1H -NMR of compound **5b**. The spectra can be opened (MestreNova file) as is not an image (all spectra in the supporting information can be checked in the same way) and studied in detail in case the referee still doubts about the reliability of the NMRs. In addition, if required we can provide all FIDs.

- The ^{13}C NMR of some products is rather diluted (e.g. 3b, etc). Please provide a better NMR for those cases.

Referee was right and better NMRs have been provided.

- Page S59, the racemic chromatogram is cut. Please provide the same range from 0 min as for the product from the catalytic reaction

A new chromatogram has been provided.

15) In general, several typos could be found along the manuscript and SI. Please check carefully and correct when necessary.

The supporting information has been revised in order to minimize typos and small mistakes and misspellings.

Comments referee 2:

Overall, this is a nice paper, but unfortunately, the reviewer is not convinced that the work is of wide enough impact to merit publication in Nature Communications.

We would like to thank the reviewer for his/her careful reading of our manuscript, for his/her appreciation considering it as a “nice paper” and for his/her comments which we address below, in which we have tried to convince him/her about the merit of our study to be published in this journal.

The authors claimed: “Anion-binding processes are well-known for their crucial role in molecular recognition. This type of phenomenon has only recently been utilized for catalysis.” This kind of interaction has been used in ion-pair catalysis. Therefore, although it is new for desymmetrization, it is just one example, not an extensive scope. Can it be transferred to other symmetrical species? Can it be done with other anions?

As the referee said the presented study only involves a desymmetrization reaction though an enantioselective deprotonation. A full study on that respect has been presented, including of course the scope of the reaction. We believe that the protocol here established might be extensible to other systems that involve a deprotonation event in a similar way that mediated by a chiral base overall. However, we have not tested those other reactions as they are out of the purpose of the presented work. Representing a completely new study. Of course, it would be highly interesting to study new reactions and the possibility to understand if other anions can potentially be recognized by these catalysts and for sure we will consider it in the future.

In Scheme 1, the authors introduce the concept of a "chiral close ion-pair." However, the description of this interaction as a "close ion-pair" may not be entirely accurate. The positioning of charges and the distance between them raise concerns, as there is a complex web of interactions, including hydrogen bonds, dipole-dipole forces, and various non-covalent interactions (NCIs), which complicates the oversimplified label of "ion-pair." Additionally, the anion of the sulfonium salt in this model is not clearly defined, and its role in the computational study remains unclear.

We thank the reviewer for his/her comment. Ion-pair is one of the most relevant interactions since a bicarbonate anion and a sulfonium cation, together with the catalyst, form the complex. However, we cannot discard other non-covalent interactions (NCIs) that are occurring simultaneously, as the reviewer said. A throughout theoretical study investigating these interactions in detail has been carried out at the request of the reviewer in one of the following comments (see below).

On the other hand, it is well-known that “asymmetric ion-pairing catalysis” is the term used for this kind of reactions; see e.g.:

<https://doi.org/10.1002/anie.201205449>

<https://doi.org/10.1021/jacs.2c04759>

<https://doi.org/10.1039/C1CS15200A>

<https://doi.org/10.1021/jacs.2c08752>

<https://doi.org/10.1039/C8CC05311A>

<https://doi.org/10.1002/chem.201803752>

García Mancheño, O., Ed. Anion-Binding Catalysis; Wiley-VCH: Weinheim, Germany, 2021.

Since the term “close ion-pair” is not strictly accurate, we have clarified this point in the manuscript and has been substituted by “ion pair” in Scheme 1.

“An exhaustive analysis to characterize bonding properties in this complex has been carried out using different computational techniques (NCI[XXX], EDA-NCI[XXX], QTAIM[XXX], see Supporting Information). They have shown that the main stabilizing interaction is electrostatic, originated by the positive charge of the sulfonium cation and the bicarbonate, although other non-covalent interactions (NCIs), such as hydrogen bonds or π - π stacking, play also a non-negligible role.”

The approach adopted in this study follows a somewhat traditional sequence. A more contemporary approach would involve starting with a computational study to gain a thorough understanding of the mechanism, followed by experimental validation, and then proposing further computational investigations for subsequent validation.

We agree with the reviewer, and indeed we also believe that adopting a correct computational strategy is essential for the comprehension of the mechanisms in catalysis. In this work we have carried out the experiments and the simulations in parallel and we have fed each other from the results obtained during the project development. Some computational exploration was initially carried out while the results of the first reactions were obtained. Then, just after the optimization of the reaction conditions and the scope studies, simulations concerning the mechanisms were performed. Finally, further refinements of the simulations and additional experiments were done in parallel. During all the process, we discussed and compared between experimental and theoretical results. At some moments of the study the simulations were ahead guiding experiments and at other moments it was the other way around.

The reviewer was probably confused after the order in which we presented the results in the manuscript. In section “Mechanism proposal” we detail the experiments and simulations carried out to understand the origins of the enantioinduction and the mechanism in the desymmetrization process; we present them in a comprehensive way such as the reader follows the reasoning and results.

We believe that the revised version of the manuscript highlights the importance of the joint theoretical-experimental approach here adopted.

One notable omission is whether the solvent was considered in the optimization process, which can significantly impact reaction pathways and product outcomes. It is not clear from “including solvent effects (chloroform) using the implicit PCM model in all the cases”, including in the optimizations or in the SP calculations?

We thank the reviewer for pointing out this aspect. We have included solvent effects not only in the single point calculations but also in the geometry optimization. In the original version we wrote “including solvent effects (chloroform) using the implicit PCM model in all the cases”, but we have further clarified this point in the revised version with a new sentence:

“we have included solvent effects (chloroform) using the implicit PCM model⁴⁶ both in the geometry optimization and in the single point energy calculations.”

Examining Figure 2, it is evident that in the left panel, where DE=0.0 kcal/mol, the charged species are notably distant from each other. This observation raises doubts about whether the described "close ion-pair" is an accurate characterization. The assertion that it represents a pure ion-pair structure appears ambitious and speculative.

We have performed several analyses on the most stable structure after the exploration of the chemical space with CREST (shown in Fig 2(c)). In particular, we have characterized the interactions between the sulfonium cation and the anion complex formed by the catalyst bonded to the bicarbonate.

To this, we have first reoptimized the geometry at the M06-2X/6-31G(d,p) level of theory and performed a single point calculation at the at the M06-2X/6-31+G(d,p) level; in both cases including solvent effects. Over the wavefunction at the highest level of theory we performed the analyses. One of them is based on an energy decomposition analysis (EDA). Since the two fragments interact each other through non-covalent interactions (NCI), we have decided to use the EDA-NCI partitioning scheme proposed in J Chem Theory Comput 7, 633 (2011) by Mandado et al. After this analysis four terms are obtained: electrostatic, exchange, repulsion and polarization. In the structure shown in Fig.2(c) the dominant term is the electrostatic one, showing the importance of the ion pair in the stabilization of such complex.

In addition, we have performed further analyses to characterize the interaction, namely:

-Non-Covalent-Interaction using the NCIPLOT code

R.A. Boto et al, NCIPLOT4: <https://github.com/juliacontrerasgarcia/ncipLOT>

E.R. Johnson et al, J. Am. Chem. Soc. 2010, 132, 6498.

J. Contreras-García et al, J. Chem. Theory Comput. 2011, 7, 625.

and

-Quantum Theory of Atoms in Molecules (QTAIM) by Bader:

R.F.W. Bader, Chem. Rev., 1991, 91, 893

R.F.W. Bader, Atoms in Molecules: A Quantum Theory, Oxford Univ (USA), 1994

We provide all the details concerning the new calculations and the obtained results in the supporting information of the revised version. We also discuss on the kind of interactions obtained in the main manuscript.

Comparing structure I from Scheme 4 with the most stable conformer according to CREST in Figure 2, it becomes apparent that these are not identical structures. The 3D representation in Figure 2 shows the carbonate forming a parallel hydrogen bond with the squaramide, while the 2D structure in Scheme 2 exhibits a bifurcated bond. This discrepancy prompts questions regarding the mechanism of deprotonation, especially considering the distance between the sulfonium salt and the bicarbonate. A more comprehensive explanation of the pre-transition state assembly is required, as well as clarification on how to transition from the most stable conformer to TSI as presented in Table S3. This transition involves substantial reorganization of hydrogen bonds, and the change in the deprotonation site remains unexplained.

The reviewer is right, both are different structures.

In Fig.2.(c) we represent the most stable structure as found with CREST code using the GFN2-xTB level of theory in the exploration of the chemical space. However, in Scheme 4 we show another minimum in the potential energy surface where the reactant is found in a proper orientation to undergo deprotonation. Structure I, shown in Scheme 4 is the most stable isomer found when we used the more accurate DFT-M06-2X method.

We thank him/her for pointing out the difference between the structures and for asking clarification on this aspect. In the revised version we clarify this point and in the supporting information we provide both structures.

Furthermore, the notably short H-O bond lengths within the transition states seem to suggest that the bond is already formed, which warrants further discussion and clarification.

Relevant bond lengths in TS and TS' are given in Fig.3. The reviewer is right and an O-H distance of 1.174 Å is short enough to consider that a new covalent bond is being formed in the TS.

To further clarify this point, we have performed additional QTAIM calculations to verify the presence of such bond. We can see a bond critical point (BCP) characteristic of a covalent O-H bond in the TS. In the manuscript we have clarified the formation of the new H-O bond in the TS.

“The O-H distance corresponding to the proton abstraction in the TS is 1.174 Å, characteristic of a covalent bond (see Figure 3), also confirmed with the presence of a bond critical point in the QTAIM simulations (see S.I).”

The authors claim that the mechanism for the minor enantiomer resembles an E1 mechanism, unlike the major enantiomer, which shows similarities to an E2 mechanism. However, the absence of a minimum corresponding to the carbocation raises doubts. It would be prudent to perform an intrinsic reaction coordinate (IRC) analysis to confirm the reaction pathway, which should provide straightforward validation.

Regarding TS', is there any steric issue between the phenyl group of the sulfonium and the squaramide catalyst?

We have followed the reviewer suggestion and we have performed an IRC calculation to confirm the two minima that the TS connects. The pre-transition state minimum found after the IRC simulation shows a geometry slightly different to structure **I** initially computed (see figure below):

Figure. (a) Structure **I** as initially computed in the previous version (b) Optimized geometry after and IRC from the TS. Relative energies (in $\text{Kcal}\cdot\text{mol}^{-1}$) are given: ΔE electronic energy; $\Delta E(\text{ZPE})$ electronic energy corrected with ZPE; ΔG Gibbs free energy. The electronic energies have been computed at the M06-2X/6-31+G(d,p) level of theory and the corrections at the M06-2X/6-31G(d,p) level; in both cases including solvent effects.

Both minima are almost degenerated, being the original structure **I** in Fig(a) slightly more stable as computed with ΔE or $\Delta E(\text{ZPE})$. The new IRC minimum, shown at right in the figure, is however more stable as computed with ΔG . Following this last criterium, we have updated in the revised version structure **I** by the new minimum obtained after the IRC.

The results of the new simulations confirming the reactants and products are given in the supporting information of the revised version.

We thank the reviewer for his/her suggestion.

We have also included in the supporting information the bonding analysis of the new structure **I**: NCI, EDA-NCI and QTAIM.

Concerning TS', we present other views of the optimized geometry in the supporting information to clarify the question. We can observe that one of the phenyl groups of the sulfonium is pointing out the structure and the other phenyl group interacts with the $-\text{Ph}(\text{CF}_3)_2$ of the squaramide.

Given that this work is based on hydrogen bonds and weak interactions, a more comprehensive analysis and characterization of non-covalent interactions (NCIs) upon complexation are crucial to elucidate the underlying rationale for the study's findings. Additionally, if the authors wish to discuss ion-pair interactions, an analysis of the nature of these interactions through tools like Quantum Theory of Atoms in Molecules (QTAIM) and energy decomposition analysis is necessary to substantiate their claims.

We thank the referee for his/her suggestion. We have performed three different analyses: NCI, EDA and QTAIM, on relevant structures in the paper:

-the most stable structure found in the chemical space exploration as obtained with CREST

-the new structure **I**, found after the IRC calculation, i.e. the pre-transition complex before reacting through TS.

-the TS that explains the enantioselectivity.

All these analyses, already mentioned and partially explained in previous questions, have been included in the revised version of the supporting information. They provide comprehensive and complementary information on the interactions between reactant and catalyst, which are the origin behind the catalytic activity.

In Table 1, which discusses the catalyst scope, it is evident that replacing OMe with H has a substantial impact. It is essential to consider whether a computational study was conducted to elucidate the reasons behind this effect. Moreover, the choice of squaramide over thiourea in cat **2e**, which yielded the best experimental outcomes, should be explained.

We do not agree with the referee, since in Table 1 the best results are obtained for catalyst **2d**; see e.g. entry 12 with a yield of 90% and an enantiomeric ratio of 95:5. This is the reason of having chosen **2d** for the computational study.

In summary, this study raises numerous questions, making it an interesting yet incomplete piece of work. The claim of groundbreaking anion recognition may require further substantiation, and while it shows promise, it may not meet the criteria for publication in a prestigious journal like Nature.

We thank the reviewer again for considering our work interesting.

We hope that with all the new simulations and changes introduced in the manuscript following his/her suggestions, the manuscript has achieved the required completeness.

Further comments:

why? Furthermore, cat **2e** shows the best experimental outcomes with room temperature, a 90% yield, and a 95:5 enantiomeric ratio. Why choose squaramide over thiourea?

This comment is related with a previous one; we already answered before that the best catalyst is **2d** and not **2e** (see Table 1, entries 5 and 6).

There are many questions that make this work interesting but incomplete. Additionally, the idea of anion recognition may not be as groundbreaking as suggested in the introduction. This is a very promising piece of work that could be published in a high-impact factor journal, but perhaps not in Nature.

Again, this is a similar comment to a previous one already answered. The revised version is much more complete; it includes all the analysis suggested by the reviewer as well as clarification on the kind of interactions between catalyst and reactants in a full characterization study.

Reviewers' Comments:

Reviewer #1:

Remarks to the Author:

In this revised version of the manuscript, the authors have addressed most of the issues pointed out by this reviewer. However, an unsatisfactory answer to avoid further derivatization of the products has been given. Considering the high standards and broad readership of this journal, maybe a total synthesis is not necessary but at least a few additional derivatization (e.g. epoxidation/aziridination; 2+2 cycloaddition, pyrazole synthesis, etc) to show the applicability of the method can be expected. To sum up, the authors have substantially improved the revised work and, therefore, it can be now more strongly recommended for publication. However, 2-3 further illustrative derivatization of the product (besides the given carbonyl reduction and Michael addition to the α,β -unsaturated unit) should be still provided.

Reviewer #2:

Remarks to the Author:

File attached.

The authors have made improvements to the paper based on reviewers' comments, for which we thank them. However, some minor issues persist:

- The first issue pertains to the terminology "ion-pair catalysis," which has been misused by experimentalists for many years. While experimentalists commonly refer to this type of catalysis as ion-pair, it is important to note that this terminology may not accurately depict the true nature of the interaction. In a recent paper, the different interactions and nature of such charged systems are described, suggesting and strongly encouraging experimentalists to adopt more precise nomenclature, as proposed by several authors, including Houk, many years ago. Unfortunately, experimentalists have been slow to embrace this recommendation.
- The authors have conducted an energy decomposition analysis (EDA); however, the total electrostatic value obtained (-78.2 kcal/mol) is very low to correspond to a strict ion-pair, as previously suggested. As the authors know, hydrogen bonds present an electrostatic term that is more likely to match this value. Given the considerable distance between the positive and negative charges, I doubt the authors can identify any donation or interaction between these charged atoms. I refer to a relevant paper (Dalton Trans., 2024, 53, 1322-1335) corresponding to a hydrogen-bond-assisted ion-pair at maximum. I strongly recommend adjusting the paper to reflect the true nature of the interaction and avoiding oversimplification as a simple ion-pair.
- Regarding Figure 2, it is suggested to add the level of theory into the caption for clarity.
- Regarding Table 1:

4	2c	CH ₂ Cl ₂	r.t.	90	10:90
12	2d	CHCl ₃	r.t.	90	95:5

So, basically, catalysts 2c and 2d present similar outcomes; however, just by changing a hydrogen atom to an OMe group, the experimental enantiomeric ratio (er) is completely opposite. This disparity provides valuable computational insights. Additionally, the er is reasonably good for the majority of the scaffold, which should offer further insights, but this aspect remains unexplored.

Comments Reviewer #1:

In this revised version of the manuscript, the authors have addressed most of the issues pointed out by this reviewer. However, an unsatisfactory answer to avoid further derivatization of the products has been given. Considering the high standards and broad readership of this journal, maybe a total synthesis is not necessary but at least a few additional derivatization (e.g. epoxidation/aziridination; 2+2 cycloaddition, pyrazole synthesis, etc) to show the applicability of the method can be expected. To sum up, the authors have substantially improved the revised work and, therefore, it can be now more strongly recommended for publication. However, 2-3 further illustrative derivatization of the product (besides the given carbonyl reduction and Michael addition to the α,β -unsaturated unit) should be still provided.

We thank the reviewer for highlighting the changes made in the revised version of the manuscript considering it substantially improved and for the positive evaluation of our work.

Following his/her suggestion, we have performed a few additional derivatizations showing the applicability of the method. To our delight we have included the required derivatizations of the obtained products. Thus, the preparation of two new interesting heterocyclic derivatives have been included in Scheme 2 and accordingly in the supporting information. With these two new derivatizations the new version of the manuscript shows four interesting functionalizations of the final product in order to demonstrate its versatility.

Comments Reviewer #2:

The authors have made improvements to the paper based on reviewers' comments, for which we thank them.

We thank the reviewer for highlighting the changes made in the revised version of the manuscript and for the positive evaluation of our work.

However, some minor issues persist:

- The first issue pertains to the terminology "ion-pair catalysis," which has been misused by experimentalists for many years. While experimentalists commonly refer to this type of catalysis as ion-pair, it is important to note that this terminology may not accurately depict the true nature of the interaction. In a recent paper, the different interactions and nature of such charged systems are described, suggesting and strongly encouraging experimentalists to adopt more precise nomenclature, as proposed by several authors, including Houk, many years ago. Unfortunately, experimentalists have been slow to embrace this recommendation.

We have changed the terminology "ion-pair catalysis" in the manuscript following the reviewer suggestion.

In the description of the catalytic cycle, we include in the revised version:

"Once I_{eq} is formed, I_a cation can coordinate to catalyst $2d$ by hydrogen bonding and forms the chiral hydrogen-bonded complex I as illustrated in Scheme 4."

And in the structural analysis we have included the following sentence:

"These facts show the great tendency of the organocatalyst to activate the bicarbonate anion forming a complex with sulfonium species."

We really understand the point of the referee. However, in the introduction of the manuscript, we would like to keep the ion-pair nomenclature, since it is the one used by previous authors that we are citing.

- The authors have conducted an energy decomposition analysis (EDA); however, the total electrostatic value obtained (-78.2 kcal/mol) is very low to correspond to a strict ion-pair, as previously suggested. As the authors know, hydrogen bonds present an electrostatic term that is more likely to match this value. Given the considerable distance between the positive and negative charges, I doubt the authors can identify any donation or interaction between these charged atoms. I refer to a relevant paper (Dalton Trans., 2024, 53, 1322-1335) corresponding to a hydrogen-bond-assisted ion-pair at maximum. I strongly recommend adjusting the paper to reflect the true nature of the interaction and avoiding oversimplification as a simple ion-pair.

We thank the reviewer for pointing out the importance of the hydrogen bonds and for the reference by Iribarren et al, where the H-bond-assisted ion-pair is theoretically investigated.

We have specified in the description of the formation of complex **I**, that the sulfonium and the bicarbonate are not directly interacting but bonded to the catalyst through intermolecular non-covalent interactions, mainly through hydrogen bonds.

- Regarding Figure 2, it is suggested to add the level of theory into the caption for clarity.

We have included the level of theory in the caption of Figure 2, following the reviewer suggestion.

• Regarding Table 1:

4	2c	CH ₂ Cl ₂	r.t.	90	10:90
12	2d	CHCl ₃	r.t.	90	95:5

So, basically, catalysts 2c and 2d present similar outcomes; however, just by changing a hydrogen atom to an OMe group, the experimental enantiomeric ratio (er) is completely opposite. This disparity provides valuable computational insights. Additionally, the er is reasonably good for the majority of the scaffold, which should offer further insights, but this aspect remains unexplored.

We appreciate referee comment. The referee is absolutely right. However, this data shown in entry 4 of table 1 is simply a mistake. Now we can appreciate that both catalysts lead to the same major enantiomer as both catalyst **2c** and **2d** contain the same pseudo-enantiomer of the chiral fragment. We really want to apologise for the mistake and the confusion caused. The new version of the manuscript finally shows the correct data.

Reviewers' Comments:

Reviewer #2:

Remarks to the Author:

All comments have been addressed by the authors, rendering the paper ready for publication. Thank you.